# FavEN: Fast Audio-Visual Embodied Navigation in 3D Environments

## Abstract

Achieving fast audio-visual embodied navigation in 3D environments is still a challenging problem. Existing methods typically rely on separate audio and visual data processing merged in late stages, leading to suboptimal path planning and increased time to locate targets. In this paper, we introduce FavEN, a novel transformer and mamba architecture that combines audio and visual data into *early fusion* tokens. These tokens are passed through the entire network from the initial layer on and cross-attend to both data modalities. The effect of our early fusion approach is that the network can correlate information from the two data modalities from the get-go, which vastly improves its downstream navigation performance. We demonstrate this empirically through experimental results on the Replica and Matterport3D benchmarks. Furthermore, for the first time, we demonstrate the effectiveness of early fusion in improving the path search speed of audio-visual embodied navigation systems in real-world settings. Across various benchmarks, in comparison to previous approaches, FavEN reduces the search time by 93.6% and improves the SPL metrics by 10.4 and 6.5 on heard and unheard sounds.

## 1 Introduction

Embodied navigation requires fast and accurate actions, which utilizes autonomous agents to interpret complex environments and make rapid decisions using integrated sensory data (Zhu et al., 2016; Savinov et al., 2018; Chaplot et al., 2020; Shah et al., 2021; Ahn et al., 2022; Mezghani et al., 2022). However, locating a source of sound in a real-world environment is hard, traditional audio-visual embodied navigation methods (Chen et al., 2020; 2021; 2022) often face inefficiencies due to separate processing for audio and visual inputs. For instance, an agent searching for a ringing phone in a cluttered room can experience delays and unnecessary actions if the audio and visual data are not learned effectively. Therefore, previous approaches suffer from high latency and cannot be deployed in the real-time settings, as shown in Figure 1.

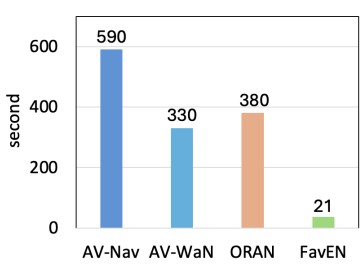

Figure 1: Search time (second) comparison with state-of-the-art methods on audio-visual embodied navigation.

Current audio-visual embodied navigation methods (Chen et al., 2020; 2021; Gan et al., 2020b; Chen et al., 2023) primarily utilize reinforcement learning (RL) strategies that treat audio and visual inputs independently or fuse them at later stages. This separation can lead to inefficient navigation paths, higher interaction costs, and extended time to complete tasks, which are practical limitations in scenarios demanding swift responses. To solve this problem, we introduce an early fusion technique in the processing pipeline, our model integrates audio and visual data right from the initial layers. This not only facilitates a deeper and more immediate understanding of the environment but also significantly reduces the number of actions and the length of search paths required to locate the target. The interaction between audio and visual features is enhanced, enabling the agent to make more informed decisions quickly. Early fusion is used to enhance model performance by preserving and utilizing the temporal and spatial correlations that naturally exist between audio and visual inputs from the outset. In our work, we aim to explore the potential of transformers for early fusion by integrating multi-modal data at the beginning of the processing pipeline, which has been largely unexplored in the context of embodied navigation.

In this paper, we introduce a novel *early fusion* architecture for Fast Audio-Visual Embodied Navigation in 3D environments (FAVEN). We first implement learnable fusion tokens within each self-attention transformer block of dedicated audio and visual encoders. These tokens serve to aggregate and refine modality-specific information throughout the network layers. To fully learn the jointly correlated information between the audio and visual data, we include cross-attention blocks to facilitate cross-modal interactions between the fusion tokens and patches from both modalities during the same forward pass. Furthermore, we introduce the novel mamba-based fusion blocks within our audio-visual navigation architecture to increase the search efficiency. This architecture ensures that the cross-modal integration of audio and visual data is not only preliminary but also deeply intertwined, allowing for a more nuanced understanding of the environment for navigation.

We validate our approach by comprehensively evaluating two established benchmarks in the field: Replica (Straub et al., 2019) and Matterport3D (Chang et al., 2017). The experimental results indicate that our method surpasses previous baselines, offering significant improvements in navigation performance. We also demonstrate the effectiveness of early fusion in extension of Mamba-based fusion blocks for audio-visual navigation. Meanwhile, we showcase the advantages of our early fusion architecture in achieving audio-visual embodied navigation for real-world settings.

In summary, we list our contributions below:

- We introduce a **fast** audio-visual embodied navigation approach using learnable fusion tokens that allows joint modeling of audio and visual data at the earliest stages of processing.

- Our approach also demonstrate efficiency of learnable fusion tokens with mamba-based fusion blocks that facilitate deep, intricate interactions between audio and visual modalities.

- We thoroughly evaluate our model on two challenging 3D environment datasets, Replica and Matterport3D, demonstrating superior navigation performance over existing baselines.

- We further showcase the generalization of our early fusion architecture in audio-visual embodied navigation for a real-world case.

## 2 RELATED WORK

In this section, we review prior work in the areas of audio-visual learning, embodied navigation, and methods of multi-modal fusion, particularly early fusion techniques. Our approach builds upon these foundations but introduces novel elements that enhance performance in audio-visual navigation tasks.

**Audio-Visual Learning.** Audio-visual learning has been extensively explored in previous works (Aytar et al., 2016; Owens et al., 2016; Arandjelovic & Zisserman, 2017; Korbar et al., 2018; Senocak et al., 2018; Zhao et al., 2018; 2019; Gan et al., 2020a; Morgado et al., 2020; 2021a;b; Hershey & Casey, 2001; Ephrat et al., 2018; Hu et al., 2019) to understand the correlation between two distinct modalities from videos. Early works like SoundNet (Aytar et al., 2016) and those by Owens et al. (Owens et al., 2016) have demonstrated the potential of leveraging cross-modal alignments for tasks such as audio-event localization (Tian et al., 2018; Lin et al., 2019; Wu et al., 2019; Lin & Wang, 2020), audio-visual spatialization (Morgado et al., 2018; Gao & Grauman, 2019; Chen et al., 2020; Morgado et al., 2020), and audio-visual parsing (Tian et al., 2020; Wu & Yang, 2021; Lin et al., 2021; Mo & Tian, 2022). Notably, recent work in audio-visual navigation by Chen et al. (Chen et al., 2020; 2021; 2022) has highlighted the challenges and opportunities in integrating these modalities for robust navigation solutions. However, our main focus is to not only capture the inherent correlations between the modalities at an early stage but also significantly enhance the navigation capabilities of autonomous agents in complex 3D environments.

**Embodied Navigation.** Embodied navigation research (Zhu et al., 2016; Savinov et al., 2018; Chaplot et al., 2020; Ahn et al., 2022; Mezghani et al., 2022) primarily focuses on enabling autonomous agents to navigate through complex environments using one or more sensory modalities. Traditional approaches (Zhu et al., 2016; Chaplot et al., 2020; Shah et al., 2021) often rely on visual inputs, but recent advancements have incorporated auditory signals to provide complementary spatial and contextual information that enhances navigational decisions (Chen et al., 2020; 2021). These integrations highlight the importance of effective multi-modal processing to interpret and react to dynamic environments accurately. In contrast, our approach sets a new benchmark for audio-visual

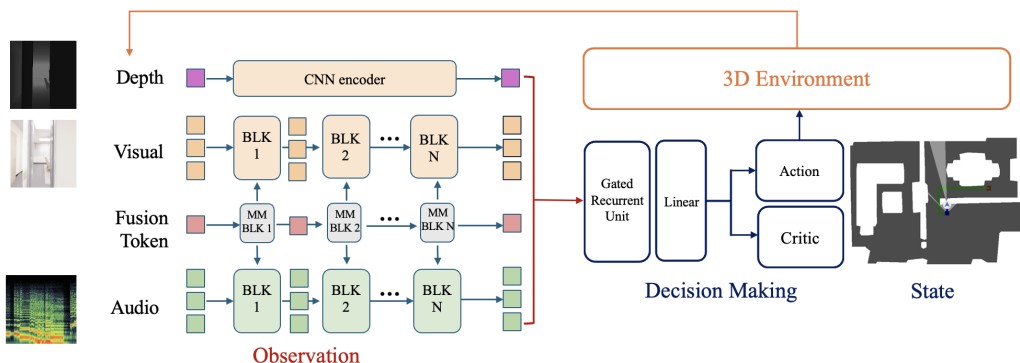

Figure 2: Illustration of the proposed fast audio-visual embodied navigation (FAvEN) architecture for embodied navigation in 3D environments. FAvEN leverages learnable fusion tokens within each self-attention transformer block (BLK 1, BLK 2, ..., BLK N) of audio and visual encoders to aggregate and refine modality-specific information throughout the network layers. Moreover, our architecture introduces multi-modal cross-attention blocks (MM BLK 1, MM BLK 2, ..., BLK N) to facilitate dense interactions between the fusion tokens and patches from both modalities during the same forward pass.

navigation tasks, opening avenues for further exploration of early fusion techniques in more complex and dynamically challenging environments.

**Multimodal Early Fusion.** The concept of early fusion, where multiple sensory inputs are integrated at the initial stages of processing, has seen varied applications and mixed results in prior studies. Owens et al. (Owens & Efros, 2018) proposed one of the early architectures for learning representations from audio-visual correspondences by concatenating features from unimodal encoders. More recent approaches have explored advanced fusion techniques, including attention-based mechanisms (Nagrani et al., 2021) and shared-weight strategies (Georgescu et al., 2023), primarily for classification tasks. However, these methods have often found mid-level fusion to outperform early fusion in tasks without a strong requirement for fine-grained multi-modal integration. In contrast to these works, we introduce a novel and faster audio-visual embodied navigation approach based on early fusion. Our approach integrates audio and visual data through learnable fusion tokens and multi-modal interaction blocks within a transformer framework.

## 3 METHOD

Given a spectrogram of audio signals, we aim to find the navigation path for localizing the sound sources in 3D environments. We propose a novel faster audio-visual embodied navigation architecture that integrates audio and visual data at early processing stages, namely FAvEN, as illustrated in Figure 2. Our observation network consists of two main modules: learnable fusion tokens for early interaction in Section 3.2 and multi-modal interaction blocks for dense fusion in Section 3.3.

### 3.1 PRELIMINARIES

In this section, we first describe the notation and revisit the audio-visual navigation problem.

**Notations.** Let $\mathcal{D} = \{(a_i, v_i, d_i) : i = 1, ..., N\}$ be a dataset of audio $a_i \in \mathbb{R}^{T \times F}$ and RGB frames $v_i \in \mathbb{R}^{T \times 3 \times H \times W}$, and depth map $d_i \in \mathbb{R}^{T \times H \times W}$ triplets. Note that $T, F$ denotes the time and frequency dimension of the audio spectrogram, respectively. For audio and images, we extracted patch embeddings from raw input via each linear projection layer, *i.e.*, $\mathbf{x}^v \in \mathbb{R}^{(V \times I) \times D}$ and $\mathbf{x}^a \in \mathbb{R}^{A \times D}$, where $I, A$ denotes the total number of patches for each video and the corresponding audio. Assume the patch resolution of each frame and audio are $P^v, P^a$, the patch-wise raw input for video and audio are formally denoted as $\mathbf{v} \in \mathbb{R}^{(V \times I) \times (3 \times P^v \times P^v)}$ and $\mathbf{a} \in \mathbb{R}^{A \times (P^a \times P^a)}$. Note that $I = H/P^v \times W/P^v, A = T/P^a \times F/P^a$. For the depth map $d_i$, we followed previous work (Chen et al., 2020; 2021) and used a CNN encoder to extract representations $\mathbf{x}^d \in \mathbb{R}^{V \times D}$ for later fusion.

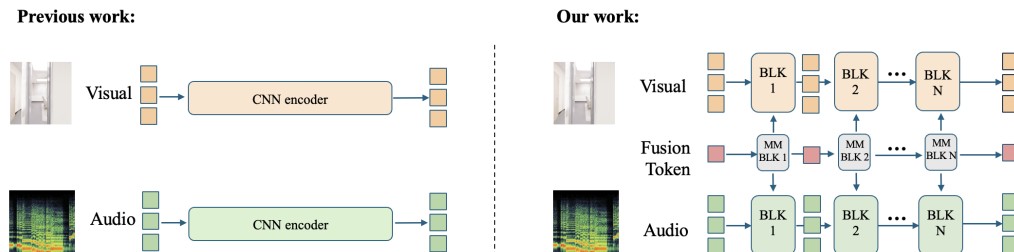

Figure 3: Comparison of the proposed fast audio-visual embodied navigation (FAVEN) architecture with previous work on embodied navigation in 3D environments. Previous audio-visual navigation methods often used separate processing streams for each modality, leading to slow response times and inefficient pathfinding. In this work, we rethink this architecture by borrowing principles from audio-visual learning to design a new architecture that integrates these modalities at the earliest stages.

**Revisit Audio-Visual Navigation.** Audio-visual navigation involves directing an agent through an environment based on inputs from both auditory and visual sensors. The primary challenge lies in effectively merging these modalities to leverage their inherent but distinct spatial and temporal characteristics, which provide complementary information critical for navigation decisions. Our approach aims to address the inefficiencies of previous methods by enhancing the interaction between audio and visual inputs, facilitating faster and more precise navigation.

**Rethinking Audio-Visual Navigation.** Current audio-visual navigation methods (Chen et al., 2020; 2021; Gan et al., 2020b; Chen et al., 2023) often utilize separate processing streams for each modality, leading to slow response times and inefficient pathfinding, especially in complex environments. State-of-the-art methods like CNNs with separate encoders for each modality often fall short in real-time applications due to the latency involved in merging the processed data, as shown in Figure 3. Our method rethinks this architecture by borrowing principles from audio-visual learning to design a system that integrates these modalities at the earliest stages, thus accelerating the decision-making process and enhancing the agent's ability to navigate dynamically changing environments efficiently.

### 3.2 LEARNABLE FUSION TOKENS FOR EARLY INTERACTION

Our architecture introduces learnable fusion tokens that serve as pivotal points for integrating audio and visual information from the very first layer of the processing pipeline. Each transformer block for audio and visual streams is equipped with a set of learnable tokens. These tokens are initialized randomly and refined through backpropagation during training. They have the capability to capture and represent key features from each modality dynamically. As data passes through each transformer block, these tokens aggregate essential modality-specific features. This aggregation is performed through attention mechanisms where the tokens learn to weigh the importance of different features within and across modalities. The updated tokens then carry forward this integrated information to subsequent layers.

During the encoding process, we apply self-attention transformers $\phi^a(\cdot), \phi^v(\cdot)$ to aggregate audio and visual features from the raw input as:

$$\phi^a(\mathbf{x}_j^a, \mathbf{X}^a, \mathbf{X}^a) = \text{Softmax}(\frac{\mathbf{x}_j^a \mathbf{X}^{a\top}}{\sqrt{D}})\mathbf{X}^a, \quad \phi^v(\mathbf{x}_j^v, \mathbf{X}^v, \mathbf{X}^v) = \text{Softmax}(\frac{\mathbf{x}_j^v \mathbf{X}^{v\top}}{\sqrt{D}})\mathbf{X}^v \quad (1)$$

where $\mathbf{X}^a = \{\mathbf{x}_j^a\}_{j=1}^{n_a}, \mathbf{X}^v = \{\mathbf{x}_j^v\}_{j=1}^{n_v}$, $n_a$ and $n_v$ denote the number of audio and visual tokens, respectively. $\mathbf{x}_j^a, \mathbf{x}_j^v \in \mathbb{R}^{1 \times D}$, $D$ is the dimension of embeddings.

In order to explicitly achieve joint audio-visual encoding across audio and visual tokens in the encoding stage, we introduce learnable fusion tokens $\{\mathbf{f}_i\}_{i=1}^{n_f}$ to aggregate audio and spatial visual features from each single-modality encoder, $\{\mathbf{x}_j^a\}_{j=1}^{n_a}, \{\mathbf{x}_j^v\}_{j=1}^{n_v}$, where $\mathbf{f}_i \in \mathbb{R}^{1 \times D}$, $n_f$ is the total number of audio-visual fusion tokens.

With learnable fusion tokens $\{\mathbf{x}_i^{av}\}_{i=1}^{n_{av}}$ and raw audio-visual representations, we first apply self-attention transformers $\phi_f^a(\cdot), \phi_f^v(\cdot)$ with context tokens to aggregate global audio and spatial visual

features from the raw input and align the features with the audio-visual context embeddings as:

$$
\{\hat{\mathbf{x}}_i^a\}_{i=1}^{n_a}, \{\hat{\mathbf{f}}_i^a\}_{i=1}^{n_f} = \{\phi_f^a(\mathbf{x}_j^{a,f}, \mathbf{X}_f^a, \mathbf{X}_f^a)\}_{j=1}^{n_a+n_f},
$$
$$
\mathbf{X}_f^a = \{\mathbf{x}_j^{a,f}\}_{j=1}^{n_a+n_f} = [\{\mathbf{x}_j^a\}_{j=1}^{n_a}; \{\mathbf{f}_i\}_{i=1}^{n_f}]
\tag{2}
$$

$$
\{\hat{\mathbf{x}}_i^v\}_{i=1}^{n_v}, \{\hat{\mathbf{f}}_i^v\}_{i=1}^{n_f} = \{\phi_f^a(\mathbf{x}_j^{v,f}, \mathbf{X}_f^v, \mathbf{X}_f^v)\}_{j=1}^{n_v+n_f},
$$
$$
\mathbf{X}_f^v = \{\mathbf{x}_j^{v,f}\}_{j=1}^{n_v+n_f} = [\{\mathbf{x}_j^v\}_{j=1}^{n_v}; \{\mathbf{f}_i\}_{i=1}^{n_f}]
\tag{3}
$$

where $[\,;\,]$ denotes the concatenation operator. $\hat{\mathbf{x}}_i^{av} \in \mathbb{R}^{1 \times D}$, and $D$ is the dimension of embeddings. Note that $\{\hat{\mathbf{f}}_i^a\}_{i=1}^{n_f}$ and $\{\hat{\mathbf{f}}_i^v\}_{i=1}^{n_f}$ will not be used as the newly updated context tokens. The self-attention operators $\phi_f^a(\cdot)$ and $\phi_f^v(\cdot)$ based on joint audio-visual encoding are formulated as:

$$
\phi_f^a(\mathbf{x}_j^{a,f}, \mathbf{X}_f^a, \mathbf{X}_f^a) = \text{Softmax}(\frac{\mathbf{x}_j^{a,f}\mathbf{X}_f^{a\top}}{\sqrt{D}})\mathbf{X}_f^a
$$
$$
\phi_f^v(\mathbf{x}_j^{v,f}, \mathbf{X}_f^v, \mathbf{X}_f^v) = \text{Softmax}(\frac{\mathbf{x}_j^{v,f}\mathbf{X}_f^{v\top}}{\sqrt{D}})\mathbf{X}_f^v
\tag{4}
$$

In each single-modality transformer block, we aggregate unimodal features with the context tokens $\{\mathbf{f}_i\}_{i=1}^{n_f}$ for joint audio-visual encoding.

### 3.3 MULTI-MODAL INTERACTION BLOCKS FOR DENSE FUSION

To enhance the integration facilitated by fusion tokens, our model includes multi-modal interaction blocks strategically positioned within the transformer framework. These blocks are designed to create dense interactions between the modality-specific patches and the fusion tokens. They employ a series of mixed attention layers where both intra- and inter-modal interactions are computed. This ensures that each token not only aggregates information from its modality but also learns from the other, creating a rich, interconnected feature space.

With the benefit of the aforementioned learnable fusion tokens, we propose a novel and explicit mechanism with multi-modal blocks for dense context interactions. Specifically, based on learnable fusion tokens $\{\mathbf{f}_i\}_{i=1}^{n_f}$ and raw audio-visual representations, we apply multi-modal blocks with fusion interaction operators $\phi_f^{av}(\cdot)$ to aggregate global audio and spatial visual features with the audio-visual fusion embeddings as:

$$
\{\hat{\mathbf{f}}_i\}_{i=1}^{n_f} = \{\phi_{ef}^{av}(\mathbf{f}_i, \mathbf{X}^{av}, \mathbf{X}^{av})\}_{i=1}^{n_f},
$$
$$
\mathbf{X}^{av} = \{\mathbf{x}_j^{av}\}_{j=1}^{n_a+n_v} = [\{\mathbf{x}_j^a\}_{j=1}^{n_a}; \{\mathbf{x}_j^v\}_{j=1}^{n_v}]
\tag{5}
$$

where $[\,;\,]$ denotes the concatenation operator. $\hat{\mathbf{f}}_i \in \mathbb{R}^{1 \times D}$, and $D$ is the dimension of embeddings. The self-attention operator $\phi_f^{av}(\cdot)$ of multi-modal blocks based on dense context interactions is formulated as:

$$
\phi_f^{av}(\mathbf{f}_i, \mathbf{X}^{av}, \mathbf{X}^{av}) = \text{Softmax}(\frac{\mathbf{f}_i\mathbf{X}^{av\top}}{\sqrt{D}})\mathbf{X}^{av}
\tag{6}
$$

In each context interaction block, we update the context tokens as $\{\hat{\mathbf{f}}_i\}_{i=1}^{n_f}$ for the input to the next interaction block to propagate cross-modal fused features. For audio and visual tokens, we use the self-attention transformers $\phi^a(\cdot), \phi^v(\cdot)$ to update audio and visual features separately for unimodal outputs defined in Eq. 1. Note that those multi-modal interaction blocks are also used for fine-tuning to improve the quality of trained audio-visual representations.

During the forward pass, audio and visual information is simultaneously processed through their respective paths. The multi-modal blocks facilitate a dynamic exchange of information, allowing the model to adjust and refine its understanding of the environment in real time. This is critical for environments where auditory and visual cues are highly dependent on each other, such as navigating through crowded or dynamically changing spaces. This methodological framework lays the groundwork for an integrated and efficient processing of audio-visual data, essential for the robust performance of navigation tasks in complex 3D environments. Subsequent sections will delve into the experimental setup, implementation details, and the comprehensive evaluation of our approach.

### 3.4 EXTENSION TO MAMBA-BASED FUSION BLOCKS

Building upon the foundational theory of State Space Models (SSMs) (Gu et al., 2022) and the Mamba (Gu & Dao, 2023) framework, we introduce the novel Mamba-based Fusion Blocks within our audio-visual navigation architecture. This extension is designed to address the computational bottlenecks typically encountered in transformer architectures, particularly those related to the quadratic complexity with respect to the length of the token sequence.

Traditional transformers exhibit quadratic computational complexity in terms of the sequence length, which becomes a limiting factor when dealing with large input sequences (e.g., 392 tokens in our experiments). To mitigate this, we integrate the Mamba framework into our fusion strategy. Mamba models, which utilize a linearized approach to handling sequences through structured global convolutions, offer a promising alternative to traditional methods by achieving linear computational complexity. Incorporating Mamba into our fusion blocks involves replacing the typical multi-head attention mechanism of transformers with a Mamba-based processing unit.

The replacement of traditional attention with Mamba-based fusion significantly reduces the computational overhead. By transforming the state-space representations into a structured global convolution format, the complexity reduces from $O(n^2)$ to $O(n)$, where $n$ is the sequence length. This reduction is crucial for scaling to larger datasets and more complex navigation tasks without compromising on processing speed or accuracy.

### 3.5 GENERALIZATION TO REAL-WORLD ENVIRONMENTS

One of the primary goals of our work is to show that FAVEN is not only effective within controlled experimental settings but also capable of generalizing to real-world environments. In this section, we outline our approach to real-world testing and demonstrate its practical applicability.

**Environment Setup.** In a real-world scenario, we have an apartment with a desk in the bedroom where a clock on a Mac computer emits periodic sounds. To achieve a realistic navigation task, we applied Blender (Denninger et al., 2023) to extract accurate camera parameters to create a spatial representation that our model could interpret. This setup aimed to mimic real-life conditions where depth cues and spatial audio play critical roles in navigating toward a sound source. In addition to audio and visual inputs, our model integrates depth information, which is crucial for navigating real environments. Specifically, we adopted Depth Anything (Yang et al., 2024) models to extract depth maps for a 3D understanding of the space, facilitating obstacle avoidance and efficient navigation. This integration showcases the model's ability to leverage and fuse multi-modal data for enhanced spatial awareness and decision-making. In this real-world scenario, the task was to navigate to the sound emitted by the clock on the computer, situated on a desk in the bedroom. Remarkably, our model successfully completed the audio-visual embodied navigation task in 21 seconds, a significant improvement over previous methods (Chen et al., 2020; 2021; 2023), which failed to reach the sound source for completing the task. The demo is provided in the supplementary material for review.

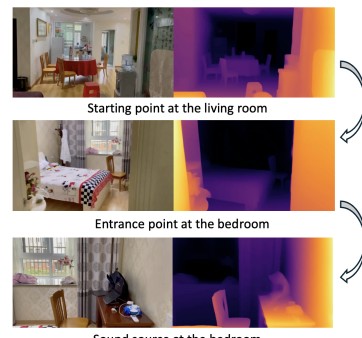

Figure 4: Illustration of the audio-visual embodied navigation for a real-world environment. We aim to find the sound source when starting from the living room in this apartment.

**Discussion.** The efficiency of FAVEN in this real-world test is attributed to its robust early fusion mechanism, which processes and integrates audio, visual, and depth cues from the start, allowing the agent to make swift and accurate navigational decisions. This is in contrast to other methods that may struggle with integrating multi-modal data, resulting in delayed responses and sub-optimal navigation. The successful navigation of FAVEN in this real-world scenario provides its potential for practical applications, particularly in environments where rapid and reliable navigation is crucial. It also highlights the model's versatility and adaptability to various real-world settings, bolstered by its capability to handle complex multi-modal sensory data effectively. This experiment serves as proof of concept that our approach can extend beyond laboratory conditions and provide substantial benefits for everyday practical use.

Table 1: **Comparison results of audio-visual navigation on Replica dataset.**

| Method | Heard | | | Unheard | | |
|---|---|---|---|---|---|---|
| | SNA ↑ | SR ↑ | SPL ↑ | SNA ↑ | SR ↑ | SPL ↑ |
| Random Agent (Chen et al., 2021) | 1.8 | 18.5 | 4.9 | 1.8 | 18.5 | 4.9 |
| Direction Follower (Chen et al., 2021) | 41.1 | 72.0 | 54.7 | 8.4 | 17.2 | 11.1 |
| Frontier Waypoints (Chen et al., 2021) | 35.2 | 63.9 | 44.0 | 5.1 | 14.8 | 6.5 |
| Supervised Waypoints (Chen et al., 2021) | 48.5 | 88.1 | 59.1 | 10.1 | 43.1 | 14.1 |
| Gan et al. (Gan et al., 2020b) | 47.9 | 83.1 | 57.6 | 5.7 | 15.7 | 7.5 |
| AV-Nav (Chen et al., 2020) | 52.7 | 94.5 | 78.2 | 16.7 | 50.9 | 34.7 |
| AV-WaN (Chen et al., 2021) | 70.7 | 98.7 | 86.6 | 27.1 | 52.8 | 34.7 |
| ORAN (Chen et al., 2023) | 70.1 | 96.7 | 84.2 | 36.5 | 60.9 | 46.7 |
| FAVEN (ours) | **76.8** | **99.7** | **94.6** | **44.5** | **67.8** | **53.2** |

## 4 EXPERIMENTS

### 4.1 EXPERIMENTAL SETUP

In this section, we describe the experimental settings used to evaluate the performance of our proposed FAVEN architecture for faster audio-visual embodied navigation in 3D environments. Our experiments are designed to demonstrate the effectiveness of our approach, which is based on early fusion and multi-modal interaction transformer blocks, in integrating audio and visual modalities.

**Datasets.** Our evaluation utilizes two major 3D environment datasets. Replica (Straub et al., 2019) is a dataset known for its high-fidelity scans of indoor environments, which provides a diverse range of audio-visual scenarios. This dataset helps in testing the robustness of our model against intricate spatial layouts with varying acoustic properties. Matterport3D (Chang et al., 2017) comprises numerous real-world spaces, offering a broader array of environmental dynamics and architectural diversity, which challenges the adaptability and scalability of our approach.

**Evaluation Metrics.** We employ three primary metrics to quantify the performance of our navigation model: 1) Success Rate (SR): measures the fraction of episodes where the agent successfully stops at the precise audio goal location. 2) Success weighted by Path Length (SPL): provides a normalized measure of success rate that accounts for the inverse of the path length, emphasizing efficiency in navigation. 3) Success weighted by Number of Actions (SNA): focuses on the number of actions taken, penalizing excessive rotations or unnecessary movements that do not contribute to successful navigation. These metrics are designed to assess not only the accuracy of the endpoint but also the efficiency and decision-making process of the navigation strategy.

**Implementation.** Our model is implemented using PyTorch (Paszke et al., 2019). We utilize an Adam (Kingma & Ba, 2014) optimizer with a learning rate of $1e - 4$ and train our models for up to 30 epochs. The audio and visual transformers consist of 6 layers each, with a hidden dimension of 512 and 8 attention heads. Early fusion tokens are introduced at each layer, allowing dynamic interaction and integration of multi-modal data throughout the training process.

### 4.2 COMPARISON TO PRIOR WORK

In this work, we propose a novel and effective framework for audio-visual embodied navigation in 3D environments. In order to demonstrate the effectiveness of the proposed FAVEN, we comprehensively compare it to the previous audio-visual embodied navigation baselines (Chen et al., 2020; 2021; Gan et al., 2020b; Chen et al., 2023).

For the Replica dataset, we report the quantitative comparison results in Table 1. As can be seen, we achieve the best results regarding all metrics for both heard and unheard settings compared to previous audio-visual navigation approaches. In particular, the proposed FAVEN superiorly outperforms ORAN (Chen et al., 2023), the current state-of-the-art audio-visual navigation baseline, by 5.1 SNA@Heard & 2.8 SR@Heard & 9.3 SPL@Heard and 6.1 SNA@Unheard & 4.8 SR@Unheard & 3.6 SPL@Unheard on two settings. Furthermore, we achieve significant performance gains compared

Table 2: **Comparison results of audio-visual navigation on Matterport3D dataset.**

| Method | Heard | | | Unheard | | |
|---|---|---|---|---|---|---|
| | SNA ↑ | SR ↑ | SPL ↑ | SNA ↑ | SR ↑ | SPL ↑ |
| Random Agent (Chen et al., 2021) | 0.8 | 9.1 | 2.1 | 0.8 | 9.1 | 2.1 |
| Direction Follower (Chen et al., 2021) | 23.8 | 41.2 | 32.3 | 10.7 | 18.0 | 13.9 |
| Frontier Waypoints (Chen et al., 2021) | 22.2 | 42.8 | 30.6 | 8.1 | 16.4 | 10.9 |
| Supervised Waypoints (Chen et al., 2021) | 16.2 | 36.2 | 21.0 | 2.9 | 8.8 | 4.1 |
| Gan et al. (Gan et al., 2020b) | 17.1 | 37.9 | 22.8 | 3.6 | 10.2 | 5.0 |
| AV-Nav (Chen et al., 2020) | 32.6 | 71.3 | 55.1 | 12.8 | 40.1 | 25.9 |
| AV-WaN (Chen et al., 2021) | 54.8 | 93.6 | 72.3 | 30.6 | 56.7 | 40.9 |
| ORAN (Chen et al., 2023) | 57.7 | 93.5 | 73.7 | 35.3 | 59.4 | 50.8 |
| FAVEN (ours) | **62.3** | **96.8** | **83.6** | **40.2** | **65.3** | **55.7** |

to AV-WaN (Chen et al., 2021), the current state-of-the-art waypoints-based baseline, which indicates the importance of incorporating cross-modal interactions from early stages in audio-visual transformer blocks as guidance for audio-visual navigation. Meanwhile, the advantage between our FAVEN and the performance of AV-Nav (Chen et al., 2020) using all data for training is the largest compared to state-of-the-art baselines. These significant improvements demonstrate the superiority of our approach in audio-visual embodied navigation.

In addition, significant gains in Matterport3D benchmark can be observed in Table 2. Compared to AV-WaN (Chen et al., 2021), the current state-of-the-art waypoints-based method, we achieve the results gains of 7.5 SNA@Heard & 3.2 SR@Heard & 11.3 SPL@Heard and 9.6 SNA@Unheard & 8.6 SR@Unheard & 14.8 SPL@Unheard on two settings. Moreover, when evaluated on the challenging Matterport3D benchmark, the proposed method still outperforms ORAN (Chen et al., 2023) by 4.6 SNA@Heard & 3.3 SR@Heard & 9.9 SPL@Heard and 4.9 SNA@Unheard & 5.9 SR@Unheard & 4.9 SPL@Unheard. We also achieve highly better results against AV-Nav (Chen et al., 2020), the late fusion network based on separate audio-visual encoders. These results demonstrate the effectiveness of our approach in learning early interaction semantics from audio and images for audio-visual navigation.

**Agent search time comparison.** A critical measure of the effectiveness of our audio-visual early fusion approach is the reduction in agent search time compared to traditional methods. The experimental results are reported in Figure 1. In our experiments, the agent equipped with our model significantly reduced the time required to locate a sound source within complex 3D environments. Specifically, our model achieved an up to 88.8% decrease in search time on the Replica dataset relative to the best-performing baseline methods. These improvements are attributed to the efficient use of early fusion tokens, which enhance the agent's ability to quickly interpret and act upon combined audio-visual cues, minimizing unnecessary navigation and expediting target location.

These comparisons not only underline the efficacy of our method but also establish a new benchmark for audio-visual navigation tasks in complex 3D settings.

## 4.3 EXPERIMENTAL ANALYSIS

In this section, we provide a detailed analysis of the experiments conducted to evaluate the performance of our Audio-Visual Early Fusion model for embodied navigation in 3D environments. The analysis is focused on understanding the contribution of the learnable fusion tokens and multi-modal interaction blocks, as well as exploring the impact of varying the number of fusion tokens and the depth of early fusion layers within the model.

**Learnable Fusion Tokens & Multi-modal Interaction Blocks & Mamba.** To validate the effectiveness of the learnable fusion tokens (LFT) and multi-modal interaction blocks (MIB), we conducted ablation studies that measure the performance degradation when each component is removed or altered. The results in Table 3 indicate that both the fusion tokens and the interaction blocks significantly contribute to the model's ability to integrate audio and visual information effectively. Models lacking fusion tokens showed a marked decrease in SR and SPL on both heard and unheard settings,

Table 3: **Ablation results of component analysis for learnable fusion tokens (LFT), multi-modal interaction blocks (MIB), and Mamba on Replica datasets.**

| LFT | MIB | Mamba | Heard | | | Unheard | | | Search Time ↓ |
|-----|-----|-------|-------|-----|-----|---------|-----|-----|---------------|
| | | | SNA ↑ | SR ↑ | SPL ↑ | SNA ↑ | SR ↑ | SPL ↑ | (s) |
| ✗ | ✗ | ✗ | 70.7 | 98.7 | 86.6 | 27.1 | 52.8 | 34.7 | 330 |
| ✗ | ✓ | ✗ | 73.7 | 99.2 | 89.3 | 29.8 | 55.2 | 37.5 | 190 |
| ✓ | ✓ | ✗ | 75.2 | 99.5 | 93.5 | 42.6 | 65.7 | 50.3 | 37 |
| ✓ | ✓ | ✓ | **76.8** | **99.7** | **94.6** | **44.5** | **67.8** | **53.2** | **21** |

Table 4: **Ablation analysis of learnable fusion tokens and the number of early fusion layers in navigation on Replica dataset.** $n_{av}, n_a, n_v$ denote the number of fusion tokens for audio-visual, audio, and visual, separately.

(a) Learnable fusion tokens.

| $n_{av}$ | $n_a$ | $n_v$ | Heard | | | Unheard | | |
|----------|-------|-------|-------|-----|-----|---------|-----|-----|
| | | | SNA ↑ | SR ↑ | SPL ↑ | SNA ↑ | SR ↑ | SPL ↑ |
| 1 | 3 | 3 | 74.1 | 99.0 | 90.6 | 37.5 | 62.9 | 47.6 |
| 3 | 3 | 3 | 74.5 | 99.4 | 91.7 | 40.7 | 63.8 | 48.5 |
| 6 | 3 | 3 | **75.2** | **99.5** | **93.5** | **42.6** | **65.7** | **50.3** |
| 12 | 3 | 3 | 75.1 | 99.4 | 93.3 | 42.3 | 65.4 | 50.1 |
| 6 | 1 | 1 | 74.6 | 99.2 | 92.5 | 41.5 | 64.2 | 49.3 |
| 6 | 6 | 6 | 74.9 | 99.3 | 92.7 | 42.1 | 65.1 | 49.8 |
| 6 | 12 | 12 | 74.3 | 99.1 | 91.2 | 39.8 | 63.1 | 48.2 |

(b) Number of early fusion layers.

| # Layers | Heard | | | Unheard | | |
|----------|-------|-----|-----|---------|-----|-----|
| | SNA ↑ | SR ↑ | SPL ↑ | SNA ↑ | SR ↑ | SPL ↑ |
| 0 | 73.7 | 99.1 | 89.3 | 29.8 | 55.2 | 37.5 |
| 1 | 74.0 | 99.2 | 90.2 | 32.3 | 58.6 | 42.7 |
| 3 | 74.3 | 99.2 | 91.1 | 37.9 | 62.4 | 47.5 |
| 6 | 74.7 | 99.3 | 91.9 | 41.3 | 63.8 | 49.2 |
| 9 | **75.2** | **99.5** | **93.5** | **42.6** | **65.7** | **50.3** |
| 12 | 75.1 | 99.4 | 93.2 | 42.3 | 65.2 | 50.1 |

confirming that these tokens play a crucial role in capturing and synthesizing modality-specific information early in the processing pipeline. Similarly, removing or simplifying the multi-modal interaction blocks resulted in poorer performance metrics, underscoring their importance in facilitating dynamic and rich inter-modal exchanges necessary for accurate navigation.

**Impact of the number of fusion tokens.** We further experimented with different configurations of fusion tokens to find the optimal number for effective performance. Initially, models were equipped with varying numbers from 1 to 10 fusion tokens per transformer block. The empirical results in Table 4a suggest a performance peak with 3 fusion tokens, beyond which additional tokens do not yield significant improvements. This observation aligns with the diminishing returns seen in models overloaded with parameters, where the complexity does not necessarily translate to better real-world performance.

**Impact of the number of early fusion layers.** The depth of early fusion—defined by the number of transformer layers in which audio and visual data are integrated from the start—also plays a critical role in performance. Our experiments varied the depth from 1 to all layers of the transformer. The results in Table 4b consistently showed that deeper integration (up to 9 layers) enhances the navigational accuracy and efficiency, as evidenced by higher SPL and SNA scores. However, extending fusion to all layers did not lead to significant gains, possibly due to overfitting on the training data or excessive entanglement of features, which might obscure useful modality-specific details. These analyses not only validate the architectural choices made in designing our model but also provide insights into the critical balance required in multi-modal learning systems. The findings from these studies will guide future improvements and refinements in the field of audio-visual navigation.

**Qualitative visualizations.** To complement our quantitative findings, we present qualitative visualizations that illustrate the navigation efficiency and decision-making process of our model, as shown in Figure 5. These visualizations include trajectory plots that compare the paths taken by our model against traditional late fusion models. For example, in a scenario within the Replica environment, our agent navigates fast toward the audio source with minimal deviation, while the baseline model could exhibit several erroneous turns and backtracks. These visualizations not only demonstrate the practical navigation superiority of our approach but also provide intuitive insights into the dynamic processing capabilities of our early fusion method.

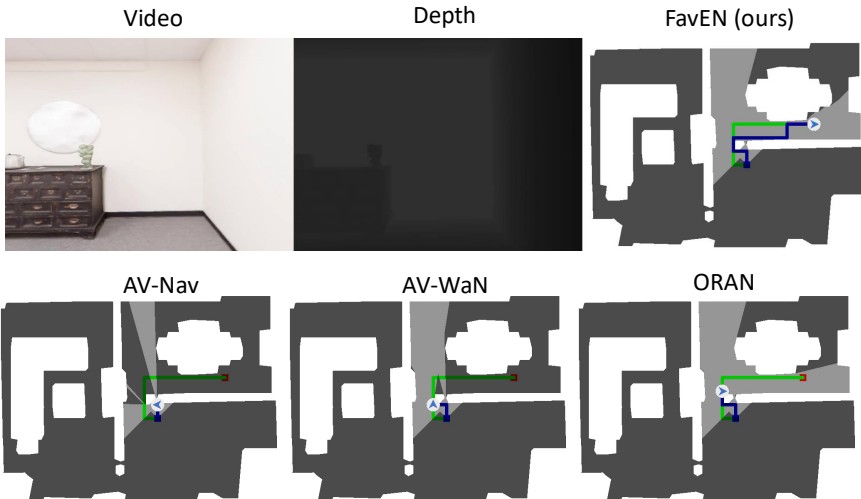

Figure 5: Qualitative visualizations of audio-visual embodied navigation. FAVEN achieves much faster results with a decent search path. The arrow in a circle denotes the direction of an agent, while the blue and green lines denote the predicted and ground-truth navigation path separately.

## 5 CONCLUSION

In this work, we present FAVEN, a novel architecture for faster audio-visual embodied navigation in 3D environments. This approach aims to enhance the integration of audio and visual information at the earliest stages of processing. Through the adoption of learnable fusion tokens and multi-modal interaction blocks within a transformer-based architecture, our model effectively captures and synthesizes the complementary modalities to improve navigation performance in complex 3D spaces. Our experimental results on the Replica and Matterport3D datasets demonstrate the superiority of our approach over traditional later fusion techniques. By implementing early fusion, our model not only achieved higher SR but also outperformed benchmarks in terms of SPL and SNA metrics. These metrics collectively highlight the efficiency and efficacy of our method in navigating accurately and economically within varied environments. Further analyses, including ablation studies on the number of fusion tokens and the depth of early fusion layers, provided valuable insights into the optimal configurations for our fusion strategy. These studies demonstrated that a balanced approach to the integration of modalities leads to more robust and adaptable navigation solutions.

**Limitations.** Despite the significant improvements in our work, we have some limitations that need further exploration. Our approach heavily relies on the quality and synchronization of audio and visual inputs. In real-world scenarios, variations in sensor quality or discrepancies in synchronization could affect the performance of our model. The effectiveness of the fusion process is contingent on the accuracy and reliability of the input data, which may not always be consistent in less controlled environments. Meanwhile, our current model integrates only audio and visual data. Extending our approach to include other modalities, such as olfactory or tactile information, could provide a more holistic sensory experience and potentially improve navigational accuracy. However, the scalability of our early fusion architecture to efficiently incorporate more modalities without a substantial increase in complexity or loss of performance is still untested. These limitations highlight the need for ongoing improvements to our model's robustness and adaptability. In future work, we can focus on optimizing the computational efficiency, enhancing the model's ability to handle variable input quality, and expanding the range of environments and modalities to which our approach can effectively adapt.

**Broader Impact.** Our findings from this work open several avenues for future work. One potential direction is the exploration of different types of fusion mechanisms that could further optimize the interaction between audio and visual cues. Additionally, extending our model to include other sensory modalities, such as olfactory or tactile information, could provide even richer environmental interactions and more nuanced navigation capabilities. Ultimately, our work contributes to the growing field of embodied AI by demonstrating that early fusion of audio and visual data can significantly enhance the operational dynamics of autonomous agents in complex, real-world settings.

ETHICS STATEMENT

We commit to the ICLR Code of Ethics and affirm that our work utilizes public datasets for experimentation. While our empirical results are largely based on publicly released datasets, we acknowledge the potential for misuse and urge the responsible application of the proposed methods with real-world data. We welcome any related discussions and feedback.

REPRODUCIBILITY STATEMENT

We provide a detailed algorithmic and experimental description in Section 4 and Appendix A & B, and we will open source the code accompanying this research upon publication.

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

---

**Algorithm 1** Algorithm for FAVEN

---

**Require:** Audio spectrograms $\mathbf{a} \in \mathbb{R}^{A \times (P^a \times P^a)}$, Video frames $\mathbf{v} \in \mathbb{R}^{(V \times I) \times (3 \times P^v \times P^v)}$
**Ensure:** Navigational actions for reaching the target
 1: Initialize audio and visual encoders $\phi^a, \phi^v$
 2: Extract audio patches $\mathbf{X}^a \leftarrow \text{Patchify}(\mathbf{a})$
 3: Extract visual patches $\mathbf{X}^v \leftarrow \text{Patchify}(\mathbf{v})$
 4: Initialize learnable fusion tokens $\{\mathbf{f}_i\}_{i=1}^{n_f}$
 5: **for** each layer $l$ in $\phi^a, \phi^v$ **do**
 6:     $\mathbf{X}^a \leftarrow \phi^a(\text{Layer}_l(\mathbf{X}^a))$
 7:     $\mathbf{X}^v \leftarrow \phi^v(\text{Layer}_l(\mathbf{X}^v))$
 8:     Update and aggregate fusion tokens:
 9:     $\{\mathbf{f}_i\} \leftarrow \text{FusionUpdate}(\mathbf{X}^a, \mathbf{X}^v, \{\mathbf{f}_i\})$
10:     Integrate audio and visual features:
11:     $\mathbf{X}^{av} \leftarrow \text{Concatenate}(\mathbf{X}^a, \mathbf{X}^v, \{\mathbf{f}_i\})$
12:     Apply multi-modal interaction:
13:     $\mathbf{X}^{av} \leftarrow \phi_f^{av}(\mathbf{X}^{av})$
14: Generate path planning decisions based on $\mathbf{X}^{av}$
15: Execute navigation actions

---

## APPENDIX

In this appendix, we provide the following material:

- addition implementation details in Section A,

- algorithm for FAVEN in Section B,

- additional experimental analyses on learnable fusion tokens in Section C,

- additional qualitative visualization results in Section D,

- a demo to show high-quality and fast navigation path generation in Section E,

- additional discussions on limitations and broader impact in Section F.

## A    IMPLEMENTATION DETAILS

In this section, we provide more implementation details. The input images are resized into a $224 \times 224$ resolution. The audio is represented by log spectrograms extracted from $3s$ of audio at a sample rate of 8000Hz. We follow the prior work (Mo & Morgado, 2022) and apply STFT to generate an input tensor of size $128 \times 128$ (128 frequency bands over 128 timesteps) using 50ms windows with a hop size of 25ms. For the audio and visual encoder, we use the single-modality MAEs (He et al., 2021; Huang et al., 2022) to initialize the visual encoder using weights pre-trained on ImageNet (Deng et al., 2009) and audio encoder pre-trained on AudioSet (Gemmeke et al., 2017). Unless other specified, the depth for multimodal blocks was set to 12. The models were trained for 100 epochs using the Adam optimizer (Kingma & Ba, 2014) with a learning rate of $1e-4$ and a batch size of 128.

## B    ALGORITHM FOR FAVEN

In this part, we provide a detailed step-by-step breakdown of the algorithm, emphasizing the dynamics of early audio-visual integration and subsequent processing. The core algorithm of FAVEN, our fast audio-Visual embodied navigation approach, involves integrating audio and visual data from the initial stages of input processing, through the application of learnable fusion tokens and multi-modal interaction blocks. Algorithm 1 outlines the process from initial data preprocessing to final decision-making, emphasizing the dynamic data integration facilitated by our multi-modal interaction blocks. The **Patchify** function refers to the extraction of patches from the raw audio and visual inputs, which are then linearly projected to match the dimensionalities required for processing by the transformer layers. The **FusionUpdate** function dynamically updates the learnable fusion tokens based on the

current states of audio and visual features, promoting an early and efficient integration. The function $\phi_f^{av}$ represents the application of mixed attention mechanisms within the multi-modal interaction blocks, allowing for complex intra- and inter-modal interactions necessary for robust navigation decision-making.

## C  ADDITIONAL EXPERIMENTAL ANALYSES

In this section, we conducted further experimental analyses to analyze two critical aspects of our system: the number of learnable fusion tokens and the depth of early fusion within the transformer layers. These studies were designed to evaluate how variations in these parameters affect the overall performance of the navigation system.

**Number of Fusion Tokens.** The configuration of fusion tokens plays a significant role in the system's ability to process and synthesize audio-visual data effectively, specifically, how many to integrate within each transformer block. We tested configurations varying from a single token to twelve tokens per modality in the transformer blocks. As illustrated in Table 4a, which provides a detailed breakdown of these configurations, we observed optimal performance with three fusion tokens per modality. This setup achieved the best balance between complexity and performance, reflected in higher SNA and SPL scores under both heard and unheard conditions. Interestingly, increasing the number of fusion tokens beyond three did not result in proportional gains in performance and, in some cases, led to a decrease in SPL and SNA scores. This suggests a point of diminishing returns, where additional tokens may introduce unnecessary complexity and redundancy, potentially leading to overfitting or less efficient training dynamics.

**Number of Early Fusion Layers.** The depth of early fusion—the number of transformer layers that integrate audio and visual data from the very beginning—is crucial for determining the system's efficacy in leveraging multimodal data for navigation. Our experiments tested various depths, from no early fusion (using separate modalities throughout all layers) to full fusion across all layers. Results, summarized in Table 4b, indicate that increasing the number of early fusion layers generally improves the system's performance, with the peak performance observed at nine layers. Beyond nine layers, we did not observe significant improvements; indeed, the performance slightly tapers off, which could be attributed to the potential for feature entanglement. When too many layers are involved in early fusion, it may lead to a blending of audio and visual cues that obscures rather than clarifies the distinct contributions of each modality, thus impeding the system's ability to make effective navigational decisions.

These findings underscore the necessity of a balanced approach to the integration of modalities in audio-visual navigation systems. While early fusion provides a powerful mechanism for leveraging multimodal data, there is a complex interplay between the number of fusion points (tokens) and the depth of their integration (layers). Optimizing these factors is crucial for designing efficient, effective systems capable of performing complex navigation tasks in dynamic environments. The insights gained from these ablation studies are invaluable for directing future research in the field. They suggest that while early fusion is beneficial, its implementation must be carefully calibrated to avoid diminishing returns and potential performance degradation. These results will guide the development of more sophisticated audio-visual navigation systems that are not only robust and effective but also computationally efficient.

## D  QUALITATIVE VISUALIZATIONS

In this section, we delve deeper into the qualitative aspects of our model's performance through comprehensive visualizations in Figure 6, 7, and 8. These visual representations are crucial for understanding how the model processes and integrates audio-visual data to make navigation decisions. They also offer insights into the practical implications of our architectural choices.

We provide detailed visualizations of navigation paths taken by the agent in various environments, illustrating the paths under both the proposed FAVEN and baseline models. These visualizations are particularly illuminative of the practical benefits of early audio-visual fusion. For instance, in scenarios featuring complex layouts with multiple potential sources of sound, the agent with FAVEN demonstrates a more direct and efficient route to the target compared to baselines.

Each visualization is accompanied by a side-by-side comparison showing the trajectory of the agent. The trajectories highlight shorter and more direct paths, reduced hesitations at decision points, and fewer incorrect turns. This directness is especially evident in cluttered or acoustically challenging environments where the integration of audio cues with visual landmarks leads to superior navigational strategies. To provide a granular view of the decision-making process, we present frame-by-frame breakdowns of certain navigation episodes. These breakdowns show the sequential processing of audio-visual data and the corresponding navigational actions taken by the agent. This step-by-step analysis helps in understanding how early integration of audio and visual data facilitates quick and accurate decision-making, improving the agent's responsiveness to dynamic changes in the environment. The qualitative visualizations not only validate the quantitative results presented earlier but also provide an intuitive understanding of why and how early fusion enhances navigation performance. These visualizations serve as a powerful tool for communicating the effectiveness of our approach to both technical and non-technical stakeholders, highlighting the practical implications of our research in real-world settings.

## E    DEMO

We provide a demo video available at an anonymous website https://fastaven.github.io/, showcasing the capability of FAVEN to generate high-quality and fast navigation paths in real-time. The demo highlights various challenging scenarios and the agent's response using our method, offering a direct view of the model's performance in dynamic settings.

## F    MORE DISCUSSIONS

In this section, we expand on the limitations discussed in the main paper and explore the broader impacts of our work, including potential societal implications and ethical considerations of deploying autonomous navigation systems in public environments. We discuss how our research could influence future developments in robotics, aiming to foster responsible innovation and use of technology.

### F.1    LIMITATIONS

While our FAVEN has demonstrated significant advancements in audio-visual embodied navigation tasks, several limitations are noted:

- **Sensitivity to Input Quality:** The performance of FAVEN heavily relies on the quality and synchronization of the input audio and visual data. In scenarios where the input data is of poor quality or improperly synchronized, the system's ability to accurately interpret and respond to environmental cues may be compromised.

- **Computational Demand:** The early fusion approach requires significant computational resources due to the simultaneous processing of audio and visual data. This may limit the deployment of our system in real-time applications on low-power devices or platforms with restricted computational capabilities.

- **Generalization Across Environments:** While tested extensively on datasets like Replica and Matterport3D, the generalizability of our method to other, perhaps more variable environments remains an open question. Environments with radically different acoustic or visual properties might pose challenges not accounted for in the current system.

- **Scalability to Additional Modalities:** The current model integrates audio and visual inputs efficiently; however, incorporating additional sensory modalities (like olfactory or tactile sensors) could complicate the fusion process, potentially reducing the system's efficiency or effectiveness.

### F.2    BROADER IMPACT

The proposed FAVEN contributes to the field of embodied AI and has several broader impacts:

- **Enhanced Accessibility:** By improving the efficiency and accuracy of navigation tasks, our method could enhance robotic applications in accessibility technology, helping people with disabilities navigate more independently in complex environments.

- **Environmental Understanding:** The advanced integration of multimodal sensory information can contribute to better environmental understanding, which is crucial for autonomous systems operating in dynamic, real-world settings. This could benefit applications ranging from autonomous vehicles to mobile robotics in disaster response scenarios.

- **Ethical Considerations:** The deployment of autonomous agents in public spaces raises important ethical considerations, including privacy and safety. The ability of these systems to interpret complex sensory data must be balanced with the need to ensure they do not inadvertently compromise individual privacy or safety.

- **Promotion of Multidisciplinary Research:** Our work demonstrates the value of interdisciplinary approaches, combining insights from robotics, artificial intelligence, and sensory processing. This can encourage further collaboration across these fields to address complex problems in novel ways.

As FAVEN continues to evolve, these discussions will guide the responsible development and application of this and similar technologies, ensuring that they not only perform effectively but also contribute positively to society.

972
973
974
975
976
977
978
979
980
981
982
983
984
985
986
987
988
989
990
991
992
993
994
995
996
997
998
999
1000
1001
1002
1003
1004
1005
1006
1007
1008
1009
1010
1011
1012
1013
1014
1015
1016
1017
1018
1019
1020
1021
1022
1023
1024
1025

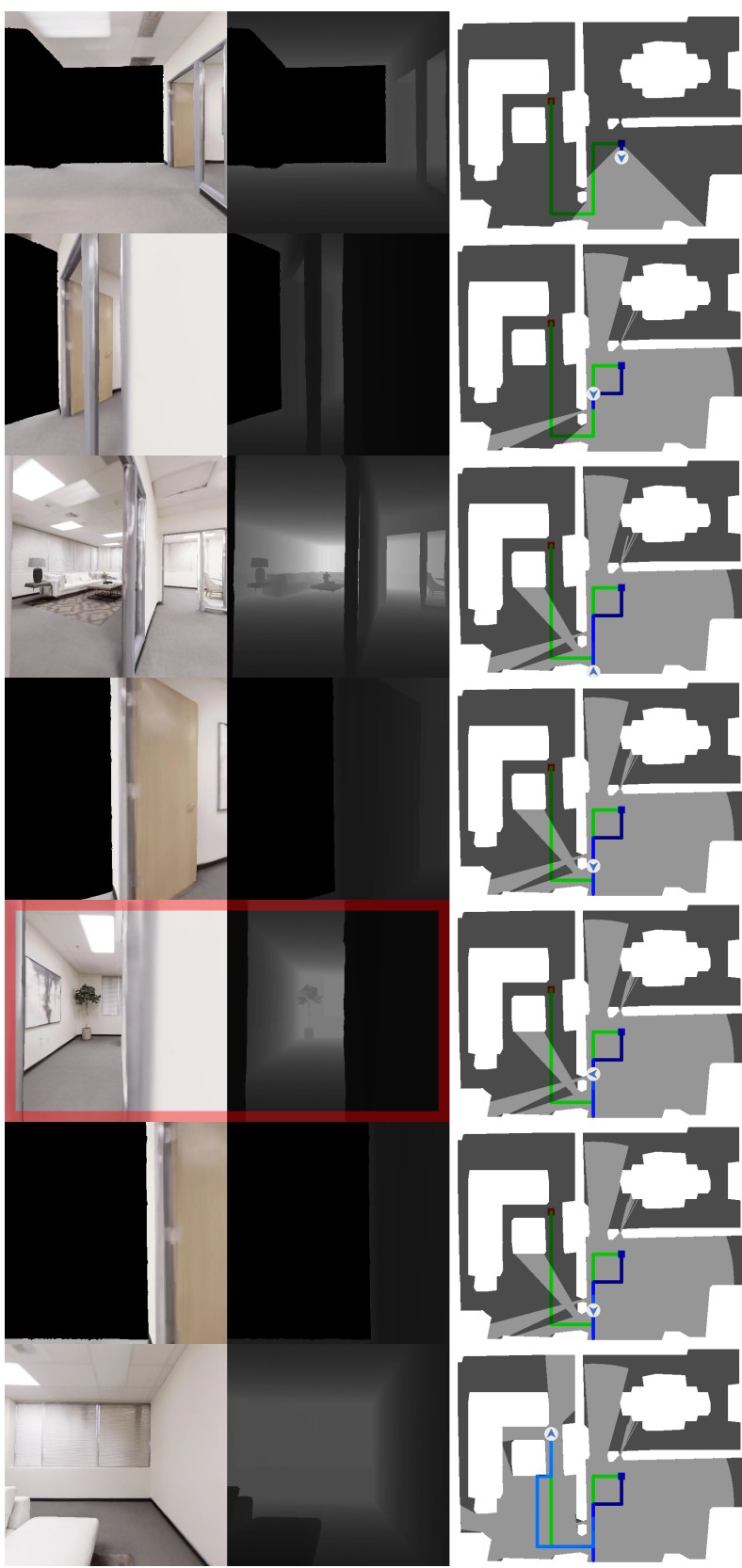

Figure 6: Qualitative visualizations of audio-visual embodied navigation. FAVEN achieves much faster results with a decent search path. The arrow in a circle denotes the direction of an agent, while the blue and green lines denote the predicted and ground-truth navigation path separately.

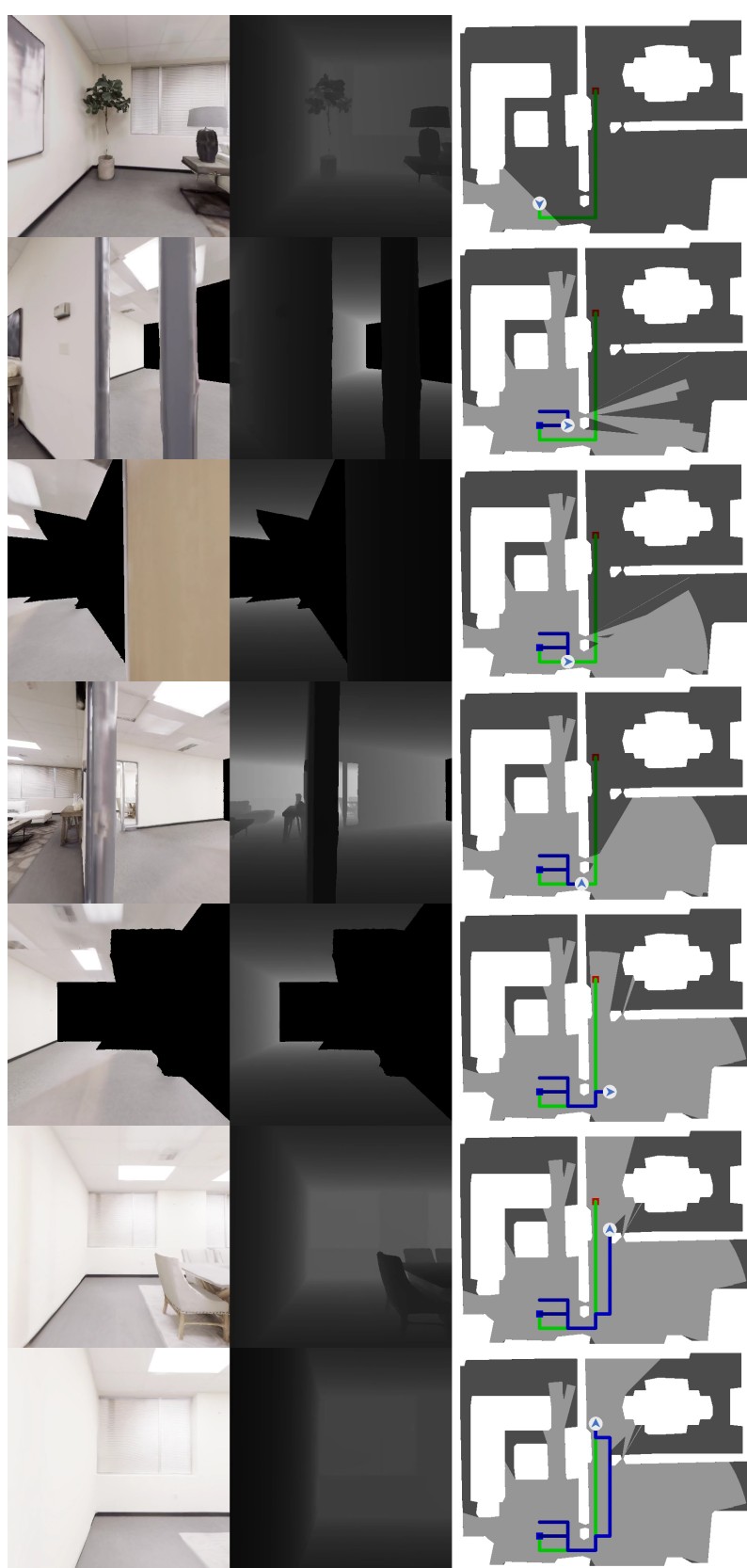

Figure 7: Qualitative visualizations of audio-visual embodied navigation. FAVEN achieves much faster results with a decent search path. The arrow in a circle denotes the direction of an agent, while the blue and green lines denote the predicted and ground-truth navigation path separately.

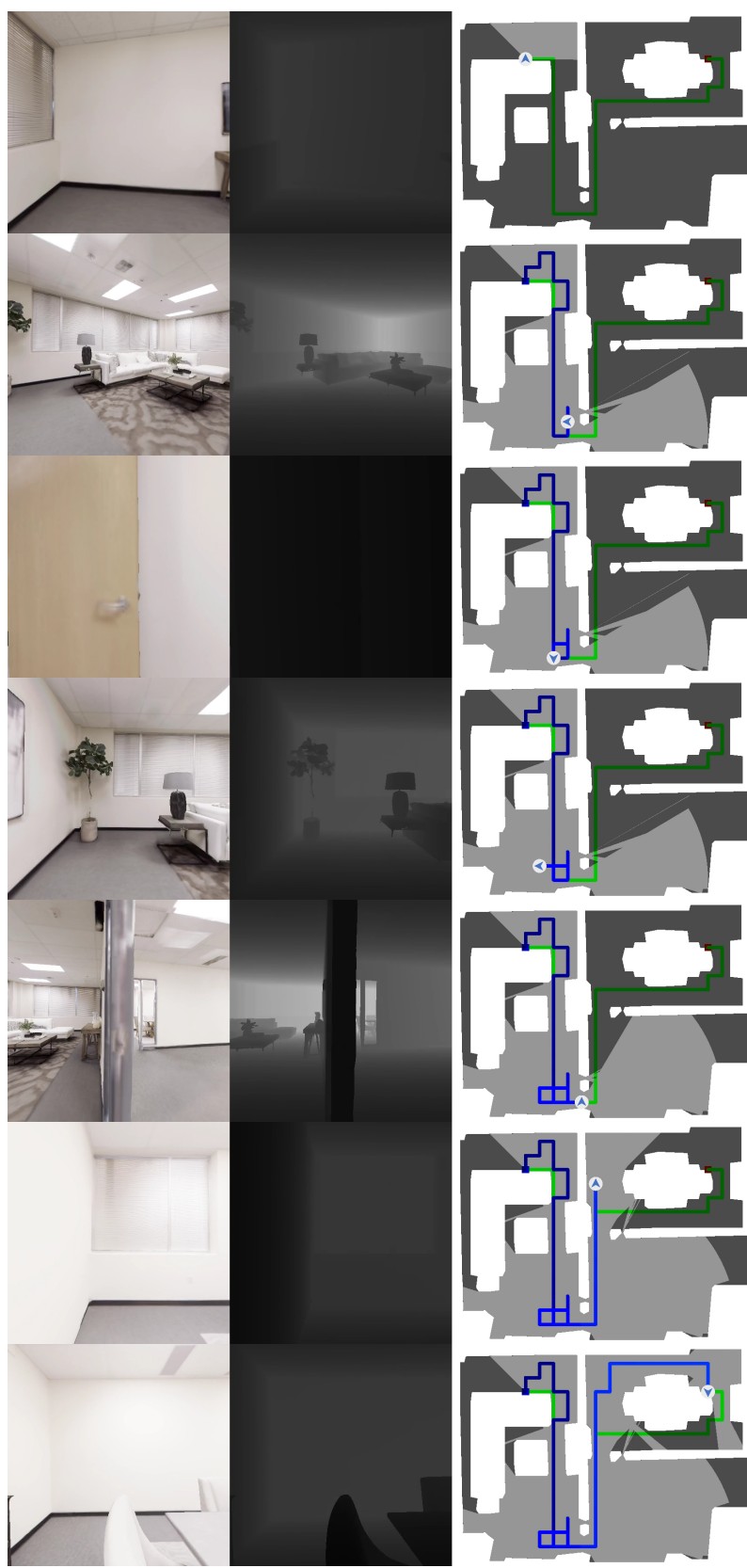

Figure 8: Qualitative visualizations of audio-visual embodied navigation. FAVEN achieves much faster results with a decent search path. The arrow in a circle denotes the direction of an agent, while the blue and green lines denote the predicted and ground-truth navigation path separately.

