# OpenReview forum: "FAVEN: Fast Audio-Visual Embodied Navigation in 3D Environments"
_ICLR.cc/2025/Conference — ICLR 2025 Conference Withdrawn Submission_

### Official Review · Reviewer_ncNK · 2024-10-20

**Soundness:** 1
**Presentation:** 2
**Contribution:** 2
**Rating:** 1
**Confidence:** 5

**Summary:**

The authors present a novel architectural approach to audio-visual navigation in three-dimensional environments. This method performs early fusion by combining audio and visual observations into tokens, thereby correlating information from both modalities to enhance the efficacy of the decision-making process. The effectiveness of this method is demonstrated by experiments conducted on Replica and Matterport3D simulation environments.

**Strengths:**

The authors introduce early fusion into audio-visual navigation by deploying a Mamba block to construct the relationship between the visual and audio embeddings. This approach has been shown to improve the success rate and efficiency of navigation to a certain extent.

**Weaknesses:**

1.	The structure of this paper is unclear and difficult to follow.
2.	The number of real-world experiments is insufficient to prove the effectiveness of the proposed method.
3.	The configurations of the real-world experiments are lacking.
4.	The comparative experiments on navigation efficiency is unclear.
5.	The contribution of this paper is weak, only the replacement of the audio and visual encoder of the AvNav, as well as the introduction of the Mamba block for early fusion.

**Questions:**

1.	The structure of this paper is unclear. For example, the authors present the real-world experiments in section 3.5, and detailed the experiments in simulated environments in section 4. It is recommended that all experiments be grouped together in a single section. In particular, it is advised that the real-world experiments be placed later in the paper, following the presentation of the simulated results.
2.	It seems that the authors conducted only one experiment in one real-world scenario (apartment). This is insufficient to prove the generalization ability of the proposed method in the real world. To achieve this, it is recommended that multiple experiments be conducted in a variety of real-world environments such as offices, classrooms, and other similar settings. Furthermore, it is also suggested that the methodology employed in the episode's production be elucidated in detail. This should include the method used to determine the robot's initial position and the target position, the distance threshold between the initial and target positions, and other relevant information.
3.	As the method proposed in this paper is for embodied audio-visual navigation, what is the configuration of the robot and the sensors (e.g., the model of the robot, the configurations of the RGB-D camera and the microphone arrays), as well as other relevant information (e.g. the sampling rate of the audio, the resolution of the images) in the real-world experiments? The video provided by the authors in the supplementary material appears to be captured by an individual using a mobile phone rather than by a camera mounted on a robot.
4.	As for the experiments on navigation efficiency, the authors said that their model achieved an up to 88.8% decrease in search time on the Replica dataset. As far as I know, the commonly used metrics on navigation efficiency are SPL, SNA and NA, as presented in Table 1 of this paper. What is the definition of the search time here and how the 88.8% calculated?
5.	The contribution of this paper is weak. In my opinion, the authors have implemented a few modifications to the observation encoder from Av-Nav. In particular, the authors replaced the audio encoder and RGB-encoder from CNN to transformer and incorporated a Mamba module for multimodal feature fusion. The article is devoid of innovation approaches to map construction, waypoint prediction, or navigation decision-making. Furthermore, the ablation experiments in this paper are insufficient. In particular, the authors conducted ablation experiments solely on Replica dataset, yet lacked experiments on Matterport3D simulated environments and real-world environments. Additionally, in the ablation study (Table 3), it is unclear which scenario was used to calculate the search time metric. Was it heard or unheard?

---

### Official Review · Reviewer_Pt9Y · 2024-11-01

**Soundness:** 2
**Presentation:** 1
**Contribution:** 2
**Rating:** 3
**Confidence:** 4

**Summary:**

This paper proposes an early fusion architecture for integrating audio and visual features to address the audio-visual embodied navigation problem. Specifically, learnable fusion tokens and cross-attention-based multi-modal interaction blocks are introduced to achieve this. The paper also attempts a Mamba-based fusion block. Experiments on the benchmarking Matterport3D dataset and Replica dataset are performed. A real-world experiment is also conducted.

**Strengths:**

1.	The paper explores an early fusion method for the visual and audio features, whose integration is crucial in addressing the audio-visual navigation task. This idea is well-motivated.
2.	Experiments on the benchmark Matterport and Replica datasets significantly outperform existing approaches, demonstrating the effectiveness of the proposed method.
3.	I am happy to see that the paper performs real-world experiments; it is a bonus to the paper.
4.	The overall idea and method are simple but effective, which is likely to impact future works on relevant problems.

**Weaknesses:**

1.	The unclear presentation of the information flow in the proposed model, especially the ordering/input/output of modules described in Section 3.2 and Section 3.3, and how they correspond to the schematics in Figure 2 and Figure 3, is very confusing.
    - I hope the authors can elaborate on the exact operations in BLKs and MM-BLKs and respond to my questions about the Equations listed in the Questions section below.
    - Other components, such as the visual encoders, GRU, Linear layer, Action and Critic modules in the system, and training objectives applied in this paper, are also unclear. I presume the authors follow some previous implementations, but it is important to clarify the entire system.
2.	The experiments presented in this paper are very shallow; this paper focuses on early fusion; hence, more in-depth experiments should be conducted on this point, including exploring other design alternatives and studying how exactly the fusion design influences the agent behavior.
    - If my understanding is correct, the proposed fusion and interaction are essentially cross-attention from visual/audio to fusion tokens and vice versa. I wonder if the authors have attempted a more efficient and deeply-bound approach by feeding all visual and audio tokens into a single multi-modal transformer for feature fusion.
    - There has no strict comparison to mid-level or late fusion based on the same pipeline.
3.	The authors highlight Mamba-based fusion blocks as one of the key contributions, but there is no information provided on the configuration of SSM or exactly how it is used to process the visual-audio tokens.
    - The authors mentioned an input sequence length of 392, which is very small, and Mamba should not show any speed advantage according to the Mamba literature and my personal experience.
    - Table 3 presents a speed advantage by comparing Mamba and softmax-attention; I suspect that the softmax-attention is a raw pytorch implementation without any CUDA optimization, such as Flash-Attention-2. In this case, the comparison is unfair to me because Mamba is nicely hardware-optimized.
    - Additionally, how to use the sequential SSM to process multimodal non-causal visual tokens is a long-lasting research problem; it is very surprising to me that using Mamba gets better results. Again, there is no explanation of the implementation in this paper. I hope the authors can clarify more.
4.	The paper claims a significant speed advantage (88.8% decrease in search time) compared to previous approaches. However, it is unclear whether the efficiency comes from the proposed fusion method (the agent runs fewer steps) or because the model runs faster. It is unclear what the architectural and inference differences are between FAVEN and the existing works. It is also unclear what the hardware is when comparing the processing speed. Overall, there is too little supporting information for the papers to make such a claim.
5.	I appreciate the real-world experiments presented in Section 3.5, however,
    - Is there any physical robot running in the real world? I am too confused by only viewing the supplemental video to understand how the entire system works. What exactly makes the control and motion?
    - I think the discussion in this section is severely overclaimed. This is only a single example, and there is no comparison to any other methods in this real-world setting.

**Questions:**

Many of my questions are included in the Weaknesses above. I hope the authors can respond to them concisely. Additionally,

1.	What is the learnable fusion tokens $\\{x^{av}\_{i}\\}^{n_{av}}\_{i=1}$ in Line 215, and how is this different from the fusion tokens $\\{f_{i}\\}^{n_{f}}_{i=1}$ in Line 211?
2.	The $x^{a,f}\_{j}$ and $x^{v,f}\_{j}$ in Equations (2) and (3) are also quite confusing, concisely, these are just the concatenation of the encoded audio/visual tokens and the fusion tokens, right?
3.	Following the above question. Where does $X^{a}$ and $X^{v}$ come from? Are they the output of Equation (1)? But in Equation (1), it seems that $X^{a}$ and $X^{v}$ are the inputs to the two intra-modal self-attention.
4.	What is $\hat{x}^{av}\_{i}$ in Line 225?
5.	Line 260: where are the “unimodal self-attention transformers” in Figure 1 and Figure 2? Are they exactly the BLKs?
6.	The Fusion Token path in Figure 1 and Figure 2 is quite misleading because audio and visual tokens are passed to the MM BLKs for interaction, but there is no arrow in the figure indicating this.
7.	The number of tokens symbols $n_{av}$, $n_{a}$, $n_{v}$, and $n_{f}$ in Section 3 and Table 4 are inconsistent; please clarify.
8.	The proposed pipeline also applies depth features, why the depth features are not considered in feature fusion? Any experiment to justify this?

Suggestions:

1.	Clearly define all symbols and notations and be consistent, e.g., $X^{av}$ and $X^{a}_{f}$ both mean a concatenated token sequence, but the latter is written as superscript and subscript.
2.	Don’t repeat simple and repetitive expressions, e.g., the softmax-attention formulation in Equations (1), (4), and (6).
3.	Use consistent names for model components, e.g., the “multi-modal blocks” and the “context interaction block”.
4.	Many descriptive statements can be made less repetitive and much more concise, e.g., Line 137-140, Line 172-175, Line 178-190, and Line 263-269 have very similar contents.
5.	Figure 3 seems like a very repetitive and not informative illustration as Figure 2. I suggest removing it.
6.	Overall, the writing of this paper looks very hasty to me. I sincerely suggest the authors carefully polish the paper and clearly all the items I listed above.

---

### Official Review · Reviewer_8ozh · 2024-11-03

**Soundness:** 3
**Presentation:** 3
**Contribution:** 2
**Rating:** 5
**Confidence:** 3

**Summary:**

In this paper the authors proposed to use learnable tokens to achieve fast audio visual navigation, where MAMBA model is used to achieve more reasonable feature learning.
The authors showcased the performance of the proposed approach in the real world scenarios.
The proposed approach is verified to be effective on 2 datasets for 3D environments navigation.

**Strengths:**

1. This paper propose a new fusion strategy to achieve more fast audio visual embodied navigation by using learnable tokens. The proposed approach is verified to be effective on diverse datastes.


2. The paper is well written and each component is verified to be effective in the corresponding ablation study.


3. The method is clearly introduced and easy to understand.

**Weaknesses:**

1. No related works in Revisit Audio Visual Navigation in Section 3.1. The authors are encouraged to add some related works in this subsection to support the claim between line 178 and line 182.

2. On line 198 the authors mentioned here, as data passes through each transformer block, these tokens aggregate essential modality-specific features. Would it be possible to qualitatively showcase the prompts perform on some specific samples (e.g., visualizations of token activations)?

3. The mamba block can not be observed in the Figure 2. The authors are suggested to revise Figure 2 accordingly to indicate where mamba block is used and how to use it.

4. Lack of comparison regarding the number of parameters. Could the authors provide the comparison of number of parameters in Table 1?

5. The authors should conduct another ablation regarding the proposed LFT compared with traditional early fusion, late fusion, cross attention fusion, etc.

6.  The LFT seems to be related to cross modal prompts. The authors are suggested to make comparison with the following work (a,b) to justify the difference and relationships. These works are also suggested to be discussed in the related work section.

a. Duan, H., Xia, Y., Mingze, Z., Tang, L., Zhu, J., & Zhao, Z. (2024). Cross-modal prompts: Adapting large pre-trained models for audio-visual downstream tasks. Advances in Neural Information Processing Systems, 36.

b. Zhai, Y., Zeng, Y., Huang, Z., Qin, Z., Jin, X., & Cao, D. (2024, March). Multi-Prompts Learning with Cross-Modal Alignment for Attribute-Based Person Re-identification. In Proceedings of the AAAI Conference on Artificial Intelligence (Vol. 38, No. 7, pp. 6979-6987).

**Questions:**

1. Could the authors include additional related works in Section 3.1 on Revisit Audio Visual Navigation to support the claims made between lines 178 and 182?

2. In line 198, the authors mention that as data passes through each transformer block, tokens aggregate modality-specific features. Could the authors qualitatively illustrate how the prompts perform on specific samples, perhaps by visualizing token activations?

3. The “mamba block” is not visible in Figure 2. Could the authors revise Figure 2 to clearly indicate where and how the mamba block is used?

4. Could the authors provide a comparison of the number of parameters in Table 1, as this would help contextualize the efficiency of the proposed model?

5. To further validate the proposed LFT, could the authors conduct an ablation comparing it with traditional fusion methods such as early fusion, late fusion, and cross-attention fusion?

6. The LFT approach seems conceptually related to cross-modal prompts. Could the authors compare their approach with other recent cross-modal prompt methods, such as those by Duan et al. (2024) and Zhai et al. (2024), to clarify differences and relationships? Additionally, would the authors consider discussing these works in the related work section?

---

### Official Review · Reviewer_1VuN · 2024-11-04

**Soundness:** 3
**Presentation:** 2
**Contribution:** 2
**Rating:** 5
**Confidence:** 4

**Summary:**

This paper proposes a novel early fusion architecture (FAVEN) for the audio-visual navigation problem. FAVEN consists of unimodal transformer blocks to process visual and audio inputs, and multimodal fusion blocks to cross-attend to information across the two modalities. Results on Matterport3D and Replica benchmarks for Audio-Visual Navigation demonstrate state-of-the-art results. Ablation studies are performed to assess the various design choices and study hyperparameters.

**Strengths:**

## Well thought-out design of architecture
The unimodal components and the fusion mechanisms are well designed and sensible. This is also the first early-fusion approach that I'm aware of for this problem space. Making this work successfully is a good value-add to the community.

## Good experiment results with SoTA
Tables 1 and 2 show comparisons against the prior SoTA methods for audio-visual navigation and FAVEN performs very well, achieving the new SoTA with a good margin.

## Good ablation studies
The ablation studies in Tables 3 and 4 addressed any initial concerns I had about the design choices made in the approach section. These are well thought out and executed.

**Weaknesses:**

## Writing clarity
* L161 - Why late fusion for depth, but early fusion for RGB and audio?
* Approach clarity
	* Eqn 1 - which blocks use this self-attention mechanism? Is it BLK 1, BLK 2, ... from Figure 3? Are there Q, K, V weight matrices applied on the embeddings before performing self-attention?
	* What is the flow of information through the architecture in Figure 3? Are equations 2 and 3 happening instead of equation 1? Does equation 5 use the outputs of equations 2 and 3?
	* L226 - Does this mean that $\hat{f}_i^a$ are discarded after each block? Are the same fusion tokens $f_i$ are used as inputs for each block?
	* Why is MAMBA needed here? Isn't 392 tokens very small compared to standard LLM applications (e.g., 100k+ tokens) where MAMBA is used?
* How is model trained? What are the loss functions employed?
* What is search time? Why is search time improvement of 88% not reflected in navigation metrics?
## Experimental limitations
* Section 3.5 - The real-world testing is very limited since it only involves one sound source in one environment for one episode. Moreover, the success / SPL / SNA metrics are not reported for all methods. L319 - 323 - the concluding claims from this experiment are very strong even though the evaluation setting is simplistic and limited.
* Missing error bars in Tables 1, 2, 3 and 4. Training policies (especially through RL) can be extremely noisy. It is good practice to train policies with multiple random seeds and report the mean and standard deviation to measure the significance of differences in performance.
* Missing comparison to other fusion mechanisms from VLM literature: This paper proposes one method of fusing information from multiple modalities. However, there are well known approaches to multimodal fusion featuring different levels of fusion (early, mid and late) in the VLM literature. Examples of models: BLIP, Unified IO, Unified IO 2, Flamingo, Chameleon, etc. These have not been qualitatively or quantitatively compared against. Note: I do realize that these have not been directly applied to the Audio-Visual navigation problem, but that does not exempt a comparison to these methods if architecture is a key contribution from the paper.
## Novelty concerns
* What is the difference between learnable fusion tokens and [REG] tokens from [Vision Transformers Need Registers](https://arxiv.org/abs/2309.16588)?

**Questions:**

I'm happy with the idea and the empirical gains of the proposed approach over prior AudioVisual Navigation methods. However, the paper's clarity is lacking, and the experiments have important limitations. There is also a novelty concern that I would like to be addressed.

---

### Note · Authors · 2024-11-17

I have read and agree with the venue's withdrawal policy on behalf of myself and my co-authors.